# A single point mutation in the *Plasmodium falciparum* FtsH1 metalloprotease confers actinonin resistance

**Christopher D Goodman\*, Taher Uddin, Natalie J Spillman, Geoffrey I McFadden**

School of BioSciences, University of Melbourne, Parkville, Australia

**Abstract** The antibiotic actinonin kills malaria parasites (*Plasmodium falciparum*) by interfering with apicoplast function. Early evidence suggested that actinonin inhibited prokaryote-like post-translational modification in the apicoplast; mimicking its activity against bacteria. However, Amberg Johnson et al. (2017) identified the metalloprotease *Tg*FtsH1 as the target of actinonin in the related parasite *Toxoplasma gondii* and implicated *P. falciparum* FtsH1 as a likely target in malaria parasites. The authors were not, however, able to recover actinonin resistant malaria parasites, leaving the specific target of actinonin uncertain. We generated actinonin resistant *P. falciparum* by in vitro selection and identified a specific sequence change in *Pf*FtsH1 associated with resistance. Introduction of this point mutation using CRISPr-Cas9 allelic replacement was sufficient to confer actinonin resistance in *P. falciparum*. Our data unequivocally identify *Pf*FtsH1 as the target of actinonin and suggests that actinonin should not be included in the highly valuable collection of 'irresistible' drugs for combatting malaria.

## Introduction

Actinonin is an anti-bacterial and anti-parasitic antibiotic derived from streptomycete bacteria (*Gordon et al., 1962*; *Wiesner et al., 2001*). In bacteria, actinonin targets peptide deformylase (PDF) (*Chen et al., 2000*), an enzyme involved in prokaryotic post-translational modification and also present in the relict plastid (apicoplast) of apicomplexan parasites. Actinonin causes defects in malaria parasite apicoplast development (*Goodman and McFadden, 2014*) and inhibits recombinantly expressed *Plasmodium falciparum* PDF (*Pf*PDF – PF3D7_0907900) in vitro (*Bracchi-Ricard et al., 2001*) at concentrations consistent with its anti-parasitic activity, all of which led to the general conclusion that actinonin targets the apicoplast-localized *Pf*PDF in malaria parasites. However, the characteristics of actinonin —particularly the rapid mode of action and the unusual kinetics of apicoplast genome loss —are at odds with how all other drugs targeting apicoplast translation impact the parasite (*Amberg-Johnson et al., 2017*; *Uddin et al., 2018*). In a search for the target of actinonin, *Amberg-Johnson et al., 2017* used the related apicomplexan *Toxoplasma gondii* and identified a point mutation in the putative metalloprotease *TgftSH1* that confers a 3.5-fold resistance to actinonin. They also showed that actinonin inhibits recombinantly expressed human malaria parasite FtsH1 (*Pf*FtsH1) in vitro at levels comparable to its antimalarial activity (*Amberg-Johnson et al., 2017*). Moreover, parasites with reduced *Pf*FtsH1 expression were more sensitive to actinonin, all of which prompted the interim conclusion that *Pf*FtsH1, rather than *Pf*PDF, might be the target of actinonin and that *Pf*FtsH1 is a potential new antimalarial target (*Amberg-Johnson et al., 2017*).

Despite repeated attempts, *Amberg-Johnson et al., 2017* were not able to generate actinonin resistant *P. falciparum* parasites. Interestingly, the residue mutated from asparagine to serine (N805S) in *Tg*FtsH1 identified as conferring actinonin resistance by Amberg-Johnson et al. (*Amberg-*

**\*For correspondence:** deang@unimelb.edu.au

**Competing interests:** The authors declare that no competing interests exist.

*Johnson et al., 2017*) is already a serine in *Pf*FtsH1 (*Table 1*), which begs the question of whether *Pf*FtsH1 is already 'resistant' to actinonin. This might mean that actinonin kills malaria parasites through a mechanism not involving *Pf*FtsH1, perhaps even inhibition of *Pf*PDF. Compounding this uncertainty is a report that *Pf*FtsH1 is localized in the mitochondrion (*Tanveer et al., 2013*), which is inconsistent with the demonstrated impact of actinonin on the malaria parasite apicoplast (*Goodman and McFadden, 2014*; *Uddin et al., 2018*). However, phenotypic evidence from *Pf*FtsH1 knockdowns and changes in post-translational processing of *Pf*FtsH1 in the absence of a functioning apicoplast (*Amberg-Johnson et al., 2017*) strongly suggest a relationship between *Pf*FtsH1 and the apicoplast. These contradictory findings may result from the of differential targeting of the various processed forms of *Pf*FtsH1 (*Tanveer et al., 2013*) or stem from the close physical and functional association between the mitochondria and apicoplast in malaria parasites (*van Dooren et al., 2005*). Given the complexity of FtsH1 processing and localization in *P. falciparum*, it is unlikley that cell biological studies alone will be able to definitively resolve the issue of whether *Pf*FtsH1 is the primary target of actinonin.

To determine if *Pf*FtsH1 is the target of actinonin, we generated *P. falciparum* parasites with robust resistance to actinonin, identified a point mutation conferring resistance, and recapitulated the resistance phenotype by introducing a single amino acid change using CRISPrCas9 genome editing.

## Results and discussion

*P. falciparum* strain D10 parasites were selected for resistance using stepwise increases in actinonin concentration. Ten million parasites were treated with 2 μM of actinonin, which resulted in no parasites being detectable in the culture by microscopy. Fresh, drug-containing media was regularly provided until parasites were again detectable by microscopy, and normal growth rate had resumed. Drug concentration was then increased two-fold and the process repeated until parasites were growing vigorously in media containing 20 μM actinonin. Both the parasite strain and selection methodology used differ from previous attempts to generate resistance (*Amberg-Johnson et al., 2017*), which may explain why we obtained resistance where others did not.

Several clones were generated from our actinonin resistant parasite line, and each showed consistent actinonin resistance, with $IC_{50}$ values 18 to 35-fold higher than the parental line (*Table 1*, *Supplementary file 1a*). The clone with the highest level of resistance showed an $IC_{50}$ of 73.3 μM actinonin (*Table 1*, *Figure 1B,C*). We genotyped four actinonin resistant clones and all retained wild type sequences of *Pf pdf*, *Pf formyl-methionyl transferase* (*Pffmt* - PF3D7_1313200), and *Pf methionyl amino peptidase* (*Pfmap* - PF3D7_0804400) suggesting that neither *Pf*PDF nor the related apicoplast post-translational protein modifying enzymes are the primary target of actinonin. Similarly, all four actinonin resistant clones retained wild type sequence for *PfRING* (PF3D7_1405700), another actinonin target candidate (*Amberg-Johnson et al., 2017*). However, each of the clones

**Table 1.** The impact of mutations in *ftsh1* on parasite resistance to actinonin.

| Parasite | FTSH1 Peptidase Motif (partial amino acid sequence) | Actinonin $IC_{50}$ (μM) |
|---|---|---|
| *Tg* FTSH1 WT (TGGT_259260) | 804 FGRDALS**N**GASSDI 811 | 14[a] |
| *Tg* FTSH1 Act[R] | 804 FGRDALS**S**GASSDI 811 | 44[a] |
| *Pf* 3D7 FTSH1 (PF3D7_1239700) | 481 FGKSETSS**G**ASSDI 494 | 1.99 (n = 1)[b] |
| *Pf* D10 FTSH1 WT | 481 FGKSETSS**G**ASSDI 494 | 2.0 ± 0.2 (n = 4) |
| *Pf* D10 FTSH1 Act[R] | 481 FGKSETSS**C**ASSDI 494 | 73.3 ± 2.7 (n = 4) |
| *Pf* D10 (apicoplast minus) | 481 FGKSETSS**G**ASSDI 494 | 43.1 ± 4.1 (n = 2)[c] |

[a]calculated from data provided in *Amberg-Johnson et al., 2017*, [b] and [c] are both consistent with previously published data (*Goodman and McFadden, 2014*; *Amberg-Johnson et al., 2017*).

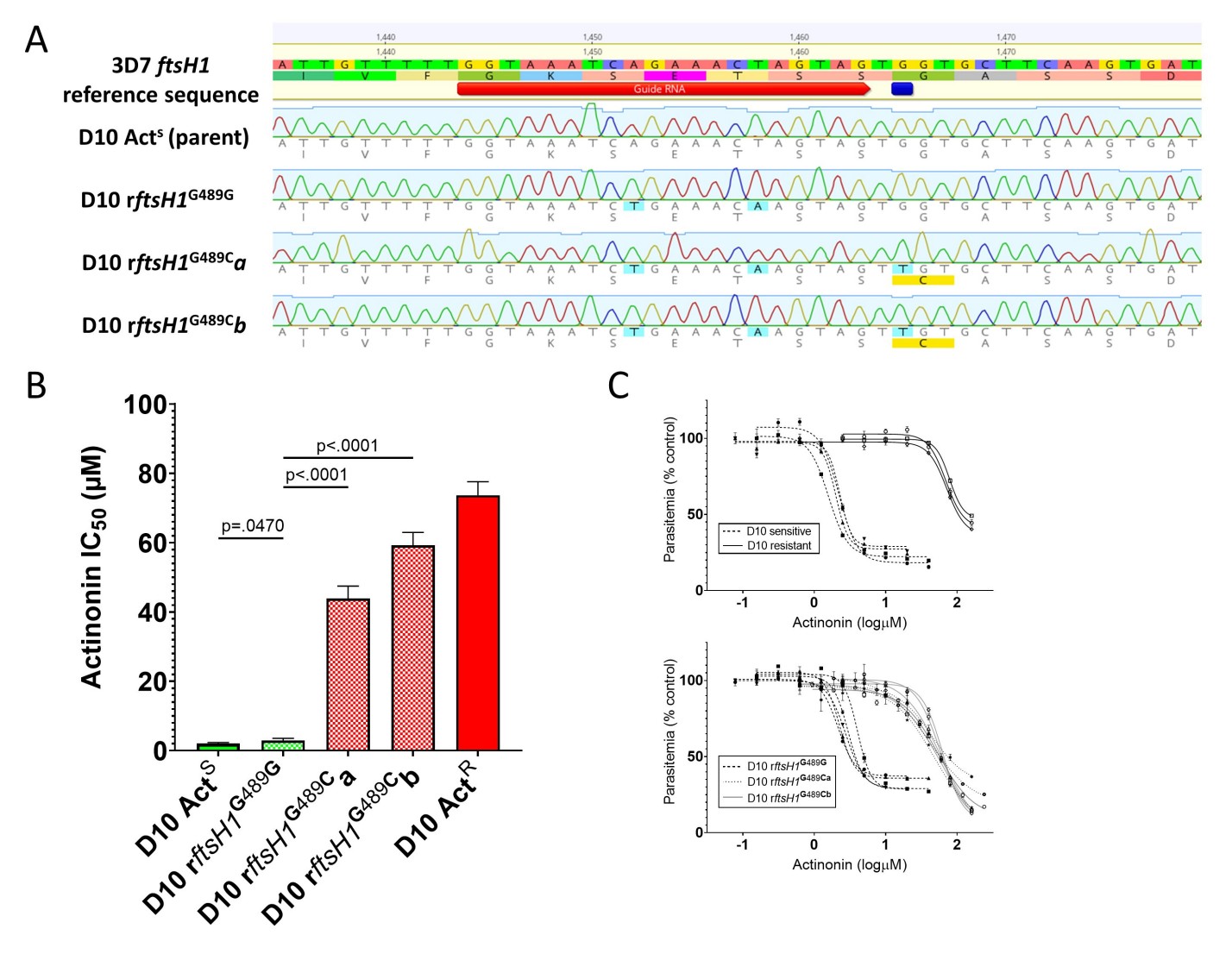

**Figure 1.** Allelic replacement in *Pfftsh1* confers actinonin resistance. (A) Genomic sequences of parasite lines. Upper line is 3D7 reference sequence with sgRNA (red arrow) and resistance mutation site (dark blue bar) marked. Bottom four lines are genomic sequence traces with shield and resistance mutations highlighted in light blue and predicted changes to amino acid sequence highlighted in yellow (B) Comparison of parasite growth inhibition ($IC_{50}$) based on the presence of the G489C mutation. (C) Dose response curves of data presented in B. Data presented are the mean of 3–5 biological replicates. Error bars represent the standard error of the mean. P values represent two-tailed, unpaired t-test. (Full statistical analysis available in *Supplementary file 1b*).

harbors a single nucleotide polymorphism in *Pfftsh1* that changed amino acid 489 (adjacent *Tg*FtsH1 N805S) from glycine to cysteine (*Table 1*, *Supplementary file 1a*), strongly implying that *Pf*FtsH1 is the primary target of actinonin.

To unequivocally confirm that *Pf*FtsH1 is the primary target of actinonin, and that the G489C mutation is sufficient to confer resistance, we used CRISPr Cas9 mutagenesis to introduce the mutation (with minimal collateral genome disruption) into the native *Pfftsh1* gene (*Figure 1A*). Accordingly, a parasite clone carrying synonymous 'shield' mutations in the *Pfftsh1* coding sequence designed to prevent ongoing Cas9 cleavage but retaining glycine 489 (rFtsH1[G489G]) remained sensitive to actinonin (*Figure 1B,C*, *Supplementary file 1b*), whereas two independent clones (rFtsH1[G489Ca/b]) modified to have the G489C mutation (in addition to the 'shield' mutations) showed actinonin resistance levels comparable to the actinonin-selected line (wtACT[R]) (*Figure 1B,C*, *Supplementary file 1b*).

Robust actinonin resistance in *P. falciparum* resulting from the G489C mutation confirms that *Pf*FtsH1 is indeed the primary target of actinonin. That the resistance levels in *Pf*FtsH1 G489C parasites are of the same order of magnitude as that seen in lines that lack an apicoplast (*Table 1*), strengthens the conclusion that *Pf*FtsH1 has a role in apicoplast biogenesis, the anomalous localization (*Tanveer et al., 2013*) notwithstanding. The greater levels of resistance achievable through prolonged selection, while modest, suggests that these lines may have acquired other mutations that compensate for reduced *Pf*FtsH1 function and/or alter secondary actinonin targets, such as the other metalloproteases present in the genome (*Amberg-Johnson et al., 2017*). Our ability to generate resistance to actinonin in a relatively small starting population of *P. falciparum* parasites means actinonin is not an 'irresistible' drug (*Cowell and Winzeler, 2018*), which tempers enthusiasm for development of *Pf*FtsH1 as an antimalarial target.

## Materials and methods

### Key resources table

| Reagent type (species) or resource | Designation | Source or reference | Identifiers | Additional information |
|---|---|---|---|---|
| Gene (*Plasmodium falciparum*) | FtsH1 | PlasmoDB (plasmodb.org) | PF3D7_1313200 | |
| Gene (*Plasmodium falciparum*) | PDF | PlasmoDB (plasmodb.org) | PF3D7_0907900 | |
| Gene (*Plasmodium falciparum*) | MAP | PlasmoDB (plasmodb.org) | PF3D7_0804400 | |
| Gene (*Plasmodium falciparum*) | FMT | PlasmoDB (plasmodb.org) | PF3D7_1239700 | |
| Gene (*Plasmodium falciparum*) | RING | PlasmoDB (plasmodb.org) | PF3D7_1405700 | |
| Strain, strain background (*Plasmodium falciparum*) | 3D7 | MR4 - BEI Resources (beiresources.org) | MRA-102 | |
| Strain, strain background (*Plasmodium falciparum*) | D10 | MR4 - BEI Resources (beiresources.org) | MRA-201 | |
| Strain, strain background (*Plasmodium falciparum*) | D10 Act$^R$ | This paper | | *Table 1* |
| Transfected construct (*Plasmodium falciparum*) | D10 r*ftsH1*$^{G489G}$ | This paper | | *Figure 1* |
| Transfected construct (*Plasmodium falciparum*) | D10 r*ftsH1*$^{G489C}$a | This paper | | *Figure 1* |
| Transfected construct (*Plasmodium falciparum*) | D10 r*ftsH1*$^{G489C}$b | This paper | | *Figure 1* |
| Software, algorithm | GraphPad Prism software | GraphPad Prism (graphpad.com) | RRID:SCR_002798 | |
| Chemical compound, drug | Actinonin | Sigma | Sigma: A6671 | |
| Chemical compound, drug | DSM-1 | Sigma | Sigma: 533304 | |

*P. falciparum* D10 were cultured according to standard protocols (*Uddin et al., 2018*; *Trager and Jensen, 1976*). Apicoplast-minus parasites were generated according to previously described methods (*Uddin et al., 2018*; *Yeh and DeRisi, 2011*). To generate actinonin resistant parasites, $10^7$ D10 parasites were treated with 2 μM actinonin (Sigma-Aldrich) and cultured until parasites began growing robustly. The actinonin concentration was then increased 2-fold and the culturing repeated until parasites grew normally at 20 μM actinonin. This selection procedure required 10 weeks of constant drug selection to recover resistant parasites and 10 months of selection to develop parasites with the highest levels of resistance. Resistant parasites were cloned by limiting dilution and retested to confirm the resistance phenotypes. Drug effects were assayed after 72 hr of drug exposure using the SYBR Green (ThermoFisher) method (*Uddin et al., 2018*; *Goodman et al., 2007*).

Genomic DNA was isolated using 200 μL of parasite culture (*Isolate II Genomic DNA kit*, Bioline). Candidate actinonin resistance genes were amplified using the primers listed in *Supplementary file 1c*. CRISPr edited FtsH1 clones were amplified using primers in *Supplementary file 1d*. Products were purified (*Isolate II PCR and Gel kit*, Bioline) and Sanger sequenced (Australian Genome Research Facility, Parkville). Alignment and analysis of sequenced genes was done using Sequencher (Gene Codes Corporation, Ann Arbor, MI USA) and Geneious Prime (www.geneious.com).

CRISPr-Cas9 mediated gene-editing utilized pAIO (*Spillman et al., 2017*) expressing Cas9 and the *Pfftsh1*-specific sgRNA 5'-GTAAATCAGAAACTAGTAG-3' inserted according to standard protocols (*Ghorbal et al., 2014*) . Two allelic replacement vectors—pFtsH1$^{G489G}$ carrying two synonymous 'shield' mutations and pFtsH1$^{G489C}$ carrying a further G to T mutation at base 1465 (*Figure 1A*)— were created by cloning a PCR amplified segment of *Pfftsh1* into pGEM-T Easy (Promega). Quick-change XL (Clontech) was used to make sequential modifications for allelic replacement constructs. The shield mutations were introduced first and then the plasmid carrying the confirmed shield mutations was modified to also include the putative resistance mutation (G1465T). All constructs were confirmed by sequencing.

Each allelic replacement vector was linearized by digestion with *Eco*RI and co-transfected with pAIO-*Pfftsh1* using standard transfection methods (*Waller et al., 2000*). Transfected parasites were selected by including 10 μM DSM-1 (Sigma-Aldrich) in the culture media for 14 days (rFtsH1$^{G489G}$ and rFtsH1$^{G489Ca}$) or 7 days (rFtsH1$^{G489Cb}$) followed by 10–14 days of culture without drug until parasites grew normally in culture. Parasites were cloned by limiting dilution and three to five clones of each line were screened for actinonin sensitivity and successful modification of the *Pfftsh1* allele. All clones from rFTSH1$^{G489G}$ and rFtsH1$^{G489Ca}$ had the expected gene modifications while only one of five clones from rFtsH1$^{G489Cb}$ did. Actinonin sensitivity was correlated to the presence of the G489C mutation in all clones tested. One clone from each recombinant line was selected for complete characterization of actinonin sensitivity.

## Acknowledgements

We thank the Australian Red Cross Blood Services, Melbourne, Australia, for supplying human erythrocytes.

## Additional information

### Funding

| Funder | Grant reference number | Author |
| --- | --- | --- |
| National Health and Medical Research Council | Project Grant APP1106213 | Christopher D Goodman Geoff McFadden |
| National Health and Medical Research Council | Project Grant APP1162550 | Christopher D Goodman Geoff McFadden |
| Australian Research Council | Laureate Fellowship FL170100008 | Geoff McFadden |
| National Health and Medical Research Council | CJ Maritn Felowship APP1072217 | Natalie Jane Spillman |

The funders had no role in study design, data collection and interpretation, or the decision to submit the work for publication.

## Author contributions

Christopher D Goodman, Conceptualization, Data curation, Supervision, Investigation, Methodology, Writing - original draft; Taher Uddin, Formal analysis, Investigation, Methodology, Writing - review and editing; Natalie J Spillman, Investigation, Methodology, Writing - review and editing; Geoffrey I McFadden, Conceptualization, Supervision, Funding acquisition, Writing - review and editing

## Author ORCIDs

Christopher D Goodman (iD) https://orcid.org/0000-0002-8923-7594

## Decision letter and Author response

Decision letter https://doi.org/10.7554/eLife.58629.sa1
Author response https://doi.org/10.7554/eLife.58629.sa2

## Additional files

### Supplementary files

• Supplementary file 1. Supplementary Tables. (**a**) Screen of D10 clones for *Pfftsh1* sequence and actinonin resistance. (**b**) Descriptive statistics for growth inhibition trials in *Figure 1*. (**c**) Oligos for PCR amplification of potential actinonin targets. (**d**) Oligos for generation and sequencing of allelic replacement constructs.

• Transparent reporting form

### Data availability

All data generated or analysed during this study are included in the manuscript and supporting files.

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
