## [Decision Letter]

Thank you for submitting your article "A single point mutation in the *Plasmodium falciparum ftsh1* metalloprotease confers actinonin resistance" for consideration by *eLife*. Your article has been reviewed by four peer reviewers, including Jon Clardy as the Reviewing Editor and Reviewer #1, and the evaluation has been overseen by Gisela Storz as the Senior Editor. The following individuals involved in review of your submission have agreed to reveal their identity: Sean Prigge (Reviewer #2); Ellen Yeh (Reviewer #3).

The reviewers have discussed the reviews with one another and the Reviewing Editor has drafted this decision to help you prepare a revised submission.

Summary:

The original report revised the generally accepted model for actinonin's antimalarial activity from inhibition of a peptide deformylase to inhibition of PfFtsH1, but a drug-resistant *PfFtsh1* mutant was not reported. This report describes a drug resistant mutant in *PfFtsh1* and supporting experiments that tighten the argument. Taken together, the results establish that FtsH1 is the primary target of actinonin in malaria parasites and show that parasites can generated significant resistance to this drug (18-35x). The reviewers generally found that this report added materially to the original report and tightened up the evidence supporting the original conclusion.

Essential revisions:

1) Introduction. Please provide more background on uncertainty of FtsH1 localization. The authors cite Tanveer et al., 2013 for mitochondrial localization of FtsH1. In Amberg-Johnson et al., 2017, cleavage of FtsH1 dependent on presence of apicoplast also indicates possible apicoplast localization. [related but different reviewer] What are the authors' thoughts about FtsH1 localization? The isoprenoid supplementation in this report and experiments in Amberg-Johnson et al., 2017, link the function of FtsH1 to the apicoplast, but the reported localization is to the mitochondrion (Tanveer et al., 2013).

2) Figure 1, Please show dose-response curves for D10 ActS, D10 ActR, FtsH1^G489G^ and rFtsH1^G489Ca/b^. The data points used to derive the IC_50_ and slope of the sigmoidal curve can be informative in interpreting the drug response.

3) The authors seem to reference the pAIO-*Pfftsh1* using Waller et al., 2000, when Spillman et al., 2017, may have been meant. Does the DHOD enzyme function in pAIO to the extent that this plasmid confers DSM1 resistance? It did not look like DSM1 was used to select for pAIO in Spillman et al., 2017.

4) The authors describe the dose escalation process of generating drug resistance, but it would be nice if they provided a sense of how long it took to achieve high level resistance.

Reviewer #1:

This seems like a pretty straightforward case. Earlier studies on actinonin's activity on the malaria parasite *Plasmodium falciparum* (Pf) left the impression that actinonin's target was peptide deformylase, its target in both bacteria and mammalian mitochondria. An *eLife* paper using a different apicomplexan, *T. gondii*, argued pretty convincingly, at least for me, that the target was likely to be a different protease, a metalloprotease called PfFtsH1 found in the Pf apicoplast. However there were alternative explanations for the results, and the target of actinonin was in the strictest sense not unambiguously identified. This short report, which shares no authors in common with the earlier Amberg-Johnson *eLife* publication, shows that a single mutation in PfFtsH1 confers resistance to actinonin in Pf. I think that the arguments presented here even more firmly establish PfFtsH1 as the target. Ironically the significance of this report, which I think is sufficient for publication in *eLife*, is to unequivocally relegate actinonin to a lesser category in the long list of potential antimalarial drugs – a relegation that I believe had largely been accomplished by the identification of FtsH1 as the target in the *Toxoplasma gondii* study reported in *eLife* by Amberg-Johnson et al.

Reviewer #2:

Actinonin is a peptidomimetic natural product known to inhibit certain metalloproteases. In bacteria and plastid-containing organisms, it has been shown to inhibit the metalloprotein peptide deformylase and this was thought to be the mechanism of action in the malaria parasite *Plasmodium falciparum*. Experiments in *Toxoplasma gondii* identified a different target metalloprotease, called FtsH1, and showed that a mutation in this enzyme conferred modest resistance to actinonin. Actinonin was also shown to inhibit recombinant *P. falciparum* FtsH1, suggesting that this is the target in malaria parasites. However, it was reported that actinonin resistance mutations could not be generated in *P. falciparum*. An additional point of uncertainty was the subcellular location of FtsH1, with one report claiming mitochondrial localization and another linking FtsH1 to the apicoplast. In the current report, the authors used a slow, dose escalation method to generate actinonin resistance in *P. falciparum* parasites. Resistant parasites contained a mutation in FtsH1 and did not have mutations in peptide deformylase or other potential target proteins. To confirm the link between drug resistance and the mutation, genome editing was used to add the mutation to a wild type line, generating resistance in this line. These experiments were carefully conducted with a genome editing control to generate a synonymous mutation that did not confer actinonin resistance. Assuming FtsH1 functions in the apicoplast, the authors used isoprenoid supplementation to disrupt the apicoplast and observed a reduced sensitivity to actinonin. Taken together, the results establish that FtsH1 is the primary target of actinonin in malaria parasites and show that parasites can generated significant resistance to this drug (18-35x).

Reviewer #3:

The identification of a PfFtsH1^G489C^ variant that confers actinonin resistance strongly supports the previous work indicating PfFtsH1 as the target of actinonin. The conclusion is well-supported by allelic replacement that recapitulated the resistance phenotype that matches well with actinonin resistance in apicoplast-minus parasites.

1) Introduction. Please provide more background on uncertainty of FtsH1 localization. The authors cite Tanveer et al., 2013, for mitochondrial localization of FtsH1. In Amberg-Johnson et al., 2017, cleavage of FtsH1 dependent on presence of apicoplast also indicates possible apicoplast localization.

2) Figure 1, Please show dose-response curves for D10 ActS, D10 ActR, FtsH1^G489G^ and rFtsH1^G489Ca/b^. The data points used to derive the IC_50_ and slope of the sigmoidal curve can be informative in interpreting the drug response.

Reviewer #4:

This study confirms a previous report by Amberg-Johnson and colleagues that a point mutation in *Plasmodium falciparumftsh1* confers resistance to a peptidomimetic antibiotic, actinonin. The authors were able to generate actinonin-resistant *P. falciparum* and identified a point mutation in *Pfftsh1* from selected resistant clones by genotyping. The resistance mutation was then introduced into the *P. falciparum* genome to verify its role in actinonin resistance. This short report adds supportive information to the previous finding that actinonin may target the apicoplast-associated FtsH1, but its novelty is considered limited.

---

## [Author Response]

Essential revisions:1) Introduction. Please provide more background on uncertainty of FtsH1 localization. The authors cite Tanveer et al., 2013 for mitochondrial localization of FtsH1. In Amberg-Johnson, 2017, cleavage of FtsH1 dependent on presence of apicoplast also indicates possible apicoplast localization. [related but different reviewer] What are the authors' thoughts about FtsH1 localization? The isoprenoid supplementation in this report and experiments in Amberg-Johnson, 2017, link the function of FtsH1 to the apicoplast, but the reported localization is to the mitochondrion (Tanveer et al., 2013).

We have expanded the discussion of this contradiction in the Introduction as follows.

“Compounding this uncertainty is a report that *Pf*FtsH1 is localized in the mitochondrion (Tanveer et al., 2013), which is inconsistent with the demonstrated impact of actinonin on the malaria parasite apicoplast (Goodman and McFadden, 2014; Uddin, McFadden and Goodman, 2017). […] Given the complexity of FtsH1 processing and localization in *P. falciparum*, it is unlikely that cell biological studies alone will be able to definitively resolve the issue of whether *Pf*FtsH1 is the primary target of actinonin.”

2) Figure 1, Please show dose-response curves for D10 ActS, D10 ActR, FtsH1^G489G^ and rFtsH1^G489Ca/b^. The data points used to derive the IC_50_ and slope of the sigmoidal curve can be informative in interpreting the drug response.

These have been included as part C in this figure, as requested.

3) The authors seem to reference the pAIO-Pfftsh1 using Waller et al., 2000, when Spillman et al., 2017, may have been meant.

Thank you for pointing this out. The reference referred to the transfection method used, not the vector. We have modified this statement for clarity. It now reads

“Each allelic replacement vector was linearized by digestion with EcoRI and co-transfected with pAIO-Pfftsh1 using standard transfection methods (Ghorbal et al., 2014).”

Please note, we also added “µM” to the concentration of DSM-1.

Does the DHOD enzyme function in pAIO to the extent that this plasmid confers DSM1 resistance? It did not look like DSM1 was used to select for pAIO in Spillman et al., 2017.

While the authors of this paper did not report selection with DSM-1 in their study, they did report the presence of a functional DHOD enzyme in pAIO. From the Materials and methods in Goodman, Su and McFadden, 2007, *“*This CAM promoter‐ yDHOD‐2A‐Cas9 amplicon was inserted into the sgRNA expression vector between XbaI/XhoI, removing the hDHFR and thymidine kinase selection cassettes, and fusing Cas9 to the *P. berghei* dihydrofolate reductase/thymidylate synthase (DT) 3′ UTR (previously supporting expression of thymidine kinase). The resulting plasmid containing both Cas9 and sgRNA expression cassettes was named pAll‐In‐One (pAIO).*”* The method of vector construction and resulting drug resistance cassette were unremarkable and it is to be expected that transfection of this construct would confer DSM-1 resistance. Our results proved this to be true.

4) The authors describe the dose escalation process of generating drug resistance, but it would be nice if they provided a sense of how long it took to achieve high level resistance.

This has been included in the Materials and methods. “This selection procedure required 10 weeks of constant drug selection to recover resistant parasites and 10 months of selection to develop parasites with the highest levels of resistance.”

Please note, for clarity we have made some minor edits. These include changes to gene names to be consistent with Amberg-Johnson et al., 2017, correction of small typographical errors, and reporting the specific concentration of actinonin used in drug resistance selection rather than relating it to IC_50_.